# Antimicrobial Prescribing Practices in Dogs and Cats by Colombian Veterinarians in the City of Medellin

**DOI:** 10.3390/vetsci8050073

**Published:** 2021-04-26

**Authors:** David A. Gómez-Beltrán, David J. Schaeffer, Duncan C. Ferguson, Laura K. Monsalve, David Villar

**Affiliations:** 1Centro de Investigaciones Básicas y Aplicadas en Veterinaria Research Group (Grupo CIBAV), Facultad de Ciencias Agrarias, Universidad de Antioquia UdeA, Calle 70 No. 52-21, Medellín 050010, Colombia; david.gomezb@udea.edu.co (D.A.G.-B.); lkatherine.monsalve@udea.edu.co (L.K.M.); 2Department of Veterinary Clinical Medicine, College of Veterinary Medicine, University of Illinois, 2001 S Lincoln Avenue, Urbana, IL 61802, USA; dschaeff@uiuc.edu (D.J.S.); dcf@illinois.edu (D.C.F.)

**Keywords:** dogs, cats, antibacterial drugs, systemic therapy, survey, Colombia

## Abstract

This study surveyed the prescribing behavior of Colombian companion animal veterinarians and compared the responses to the current guidelines of the International Society for Companion Animals on Infectious Diseases (ISCAID). A convenience sample of 100 primary-care veterinary practitioners was selected from the city of Medellin. A questionnaire was designed to present hypothetical clinical scenarios regarding prescription choices for systemic antimicrobials. The numbers of veterinarians empirically prescribing a course of systemic antimicrobials for each scenario were—perioperative elective surgeries (86%), superficial pyoderma (90%), lower urinary tract disease (52%), acute hemorrhagic diarrhea (50%), and kennel cough (46%). For urinary tract disease, cultures and susceptibility testing were only performed by half of the respondents, suggesting lower diagnostic standards. In superficial pyoderma cases, cytology was performed in the following percent of cases—0% (24), 20% (30), 40% (17), 60% (11), 80% (8), and 100% (10). Antimicrobials were over-prescribed relative to emerging standard for elective surgeries (86%), kennel cough (46%), and acute hemorrhagic diarrhea (50%). Critically important antimicrobials, such as fluoroquinolones, were applied commonly for superficial pyoderma (18%), kennel cough (12%), and lower urinary tract disease in dogs (20%) and cats (26%). In conclusion, antimicrobial prescribing behavior was inconsistent with current guidelines, and antimicrobial use could be improved by appropriate diagnostic steps allowing choice of an optimal antimicrobial drug. Overall, we documented the widespread use of antimicrobials for the treatment of these four common disease conditions.

## 1. Introduction

The increase of antimicrobial resistance (AMR) in companion animals is of growing concern. The clinical response to an empirically chosen antimicrobial is not as predictable as in the past, and culture and susceptibility analyses are increasingly indicated to select appropriate antimicrobial agents. The pressing need for prudent and focused use of antimicrobial drugs has resulted in specific veterinary specialty groups’ specific guidelines. In particular, documents by the Working Group of the International Society for Companion Animals on Infectious Diseases (ISCAID) address the diagnosis and antimicrobial therapy of canine superficial bacterial folliculitis [1], diagnosis and management of bacterial urinary tract infections [2,3], the treatment of respiratory tract disease in dogs and cats [4], and acute gastroenteritis in dogs [5]. The emphasis is placed on making an accurate diagnosis, finding alternative therapies to systemic antimicrobials, and using the antimicrobials appropriate to laboratory culture and sensitivity findings. The American Veterinary Medical Association [6] also recommends using narrow-spectrum antimicrobial agents when appropriate, using antimicrobial agents important in treating human infections only after careful review and reasonable justification, as well as treating for the shortest effective period possible to minimize therapeutic exposure.

Skin problems commonly receive systemic antimicrobial therapies. A study of 683 identified canine pyoderma cases [7] reported that 64.1% of dogs with superficial pyoderma were prescribed systemic antimicrobials only, 27.7% received both topical and systemic therapies concurrently, and 4.7% received a topical product only. However, the last is the route currently recommended by experts [1]. Lower urinary tract disease is another clinical condition in which a diagnosis is difficult to reach but for which systemic antimicrobials are commonly used unnecessarily. Specifically, subclinical infections and catheterizations should not receive antimicrobial therapy [3]. A Danish retrospective study reported that when only microscopic examinations were used for diagnosis in lieu of laboratory cultures and antibiograms, 36% of suspected cases of urinary tract infections were over-prescribed [8]. Treatment of acute diarrhea in dogs is another condition where antimicrobial use is often excessive [5]. One survey in the United Kingdom [9] and another in Australia [10] found that 71% and 32% of veterinarians, respectively, prescribed antimicrobials for non-complicated gastroenteritis cases before resorting to other more recommended options, such as nutritional management and deworming, and/or probiotics, with or without antidiarrheal agents. Because most potentially enteropathogenic bacteria are also isolated from clinically healthy dogs, the American College of Veterinary Internal Medicine (ACVIM) expert group recommendation is to only use supportive therapy [5] and avoid using antimicrobials for uncomplicated cases of acute diarrhea (even hemorrhagic).

The present study surveyed Colombian veterinarians’ antimicrobial use preferences for the most common disease conditions in clinical practice affecting dogs’ and cats’ skin, urinary, respiratory, and gastrointestinal tracts. The goal was to identify where these regional choices might better conform with international standards for rational antimicrobial use and serve as a baseline for future antimicrobial use studies.

## 2. Materials and Methods

### 2.1. Sample

A convenience selection of veterinary practices that wanted to participate in the study was selected from Medellin, Colombia, S.A. The veterinarians were initially contacted by telephone before the investigators’ visits in 2020. The questionnaire was designed for first opinion or referral practices and was based on a study regarding antimicrobial prescribing behavior in dogs and cats [11]. Ethics approval was granted by the Ethical Committee (“Comité de Bioetica para la Experimentación con humanos”) of the University of Antioquia (session number 20-113-918). Participating veterinarians were assured that their names and institutions would remain confidential. The dates for the interviews were scheduled according to the veterinarian’s convenience. The questionnaire consisted of the following sections—(1) Demographics; (2) Use of guidelines for antimicrobial prescribing; (3) Common approach to hypothetical conditions in which clinical evidence suggests a diagnosis of superficial pyoderma, acute upper respiratory infection (“kennel cough”), uncomplicated lower urinary tract infection (first occurrence), acute uncomplicated diarrhea, and elective surgeries; and (4) Semi-closed questions regarding the diagnosis procedures, selection and choice of antimicrobials, and planned duration of therapy. Respondents were also asked if they would perform in-house tests (i.e., cytology) of samples and send samples for culture and susceptibility testing.

### 2.2. Statistics

Data were collected on individual sheets in a Microsoft Excel workbook assigned to one of four interviewers during the interview. Each workbook´s sheets were compiled into a new workbook and imported into SYSTAT 13.1 (SYSTAT, Inc., San Jose, CA, USA). A preliminary 1-way contingency table was created for each question and sub-question to correct spelling or other entry errors. The code for the corrections and the cleaned data were saved as SYSTAT files. Final one-way contingency tables were created for each question, and tables for sub-questions were manually combined. The full questionnaire is available as online Appendix A.

## 3. Results

### 3.1. Demographics

Demographic characteristics of the survey respondents are summarized in Table 1. Forty-five female and 55 male veterinarians represented the study population. The majority of veterinarians worked in clinics not providing around-the-clock care. The majority of the veterinarians had less than ten years in practice. Only 11% of the veterinarians followed specific guidelines using antimicrobials, with specific protocols according to disease. Most (71%) used veterinary drug handbooks, with the most common being Plumb’s Veterinary Drug Handbook (49%), to select antimicrobials and design prescription protocols. Thirty-six veterinarians considered commercial pharmaceutical literature an important resource, of whom four used it exclusively. Seventeen veterinarians (17%) knew of specific guidelines on use of antimicrobials in small animals, although 84% thought they were necessary. The level of concern regarding antimicrobial resistance was of high (58%) or moderate (36%) interest, and 6% had little or no concern. Antimicrobial choice was often influenced by the need to minimize cost to the owner, leading to preference for generic antimicrobial products. The number of veterinarians using generic drugs varied, with 36% using them in 100% of the situations, citing cost-effectiveness as the main reason.

### 3.2. Perioperative Use of Antimicrobials

Most veterinarians (86%) felt the need to use systemic antimicrobials in 100% of the cases of ovariohysterectomies and castrations, and only 8% were confident of not using them. A total of 5% of respondents used antimicrobials in 75% of cases for these procedures. Cephalexin was the most prescribed antimicrobial (52%) for 7–10 days following either procedure. Thirty-six of 100 respondents used the combination of metronidazole and spiramycin (Stomorgyl^®^ Boehringer-Ingelheim S.A, Bogota, Colombia) for 5–7 days in 100% of dental procedure cases. The remaining veterinarians used antimicrobials as follows—14% in 75% of the cases, 20% in 50% of the cases, 26% in 25% of the cases, and 4% never.

### 3.3. Specific Disease Conditions

#### 3.3.1. Superficial Pyoderma

Most participants (76%) preferred the combination of systemic and topical therapy to treat superficial pyoderma. Eleven used systemic therapy alone, and ten used only topical therapy. The type of systemic antimicrobial agent initially prescribed and their frequency is shown in Figure 1. Chlorhexidine was cited as the first choice for topical therapy (86%), with hypochlorous acid as the second choice (14%). Cytology on first-time presentations was performed in the following percent of cases—0% (24%), 20% (30%), 40% (17%), 60% (11%), 80% (8%), and 100% (10%). Although 51% of the respondents mentioned the intention to search for underlying triggers associated with pyoderma, only 34% asked pertinent questions to identify allergic disease or mentioned potential endocrinopathies or ectoparasite infestations. The preferred answer to submit samples to the laboratory for culture and sensitivity was “whenever the empirical treatment failed” (58%), followed by “owner approval of the cost” (16%). Ten veterinarians (10%) never submitted samples for culture and ten (10%) always did. The patient’s follow-up was typically done between 5–10 days of the first visit in 46% of responses.

#### 3.3.2. Upper Respiratory Tract Disease

Of the 100 veterinarians surveyed, 46% would prescribe systemic antimicrobials to a dog with a specific upper respiratory tract disease (“kennel cough”). The preferred antimicrobial agent was amoxicillin–clavulanate (11%), followed by enrofloxacin (9%), amoxicillin (7%), azithromycin (4%), doxycycline (4%), penicillin–streptomycin (4%), marbofloxacin (3%), and cephalexin (3%). The duration of treatment was typically five days. For cats, the signs of an infection of the upper respiratory tract for which systemic antimicrobials would be prescribed are in Table 2**.**

#### 3.3.3. Lower Urinary Tract Disease

Sixty-nine veterinarians (69%) prescribed antimicrobials prophylactically for dogs and cats with the urinary bladder catheterized. The main antimicrobials used were enrofloxacin (23%), ampicillin (11%), ampicillin–sulbactam (9%), marbofloxacin (%), and trimethoprim–sulfonamide (3%). The favorite method to collect urine was cystocentesis (89%). Fifty-two percent base a diagnosis for uncomplicated urinary tract infections on the presence of lower urinary tract signs (e.g., hematuria, dysuria, pollakiuria) with a concurrent urinalysis (dipstick and cytological examination of the sediment). The other veterinarians (48%) also requested bacterial culture and sensitivities regardless of it being a dog or cat. Therefore, antimicrobial selections were empirical for about half of the veterinarians and guided by culture and susceptibility testing for the other half. There were no differences in the choice of empirical antimicrobial therapy between dogs and cats; in cats, 18% of the respondents indicated the use of enrofloxacin; other drugs were ampicillin–sulbactam (6%), ampicillin (6%), marbofloxacin (5%), doxycycline (4%), trimethoprim-sulfonamides (3%), ciprofloxacin (3%), cefovecin (3%), gentamycin (2%), and amoxicillin-clavulanate (2%) Fluoroquinolones in dogs were used by 20% of the veterinarians surveyed. Duration of therapy did not differ between dogs and cats and was predominantly ten days.

#### 3.3.4. Acute Gastroenteritis

When veterinarians were asked if they would prescribe antimicrobials to treat acute diarrhea accompanied by the following signs, the responses were—dehydration (36%), hemorrhage (74%), positive by PCR culture for enteropathogenic bacteria (81%), high rectal temperature (87%), and inflammatory leucogram (68%). For hemorrhagic diarrhea, combinations of two antimicrobials were the preferred option by 50% of the participants (Figure 2). The most selected combination was metronidazole and ampicillin.

## 4. Discussion

This study’s objective was to determine the prescribing behavior of Colombian small animal veterinarians regarding systemic antimicrobials. There are no national guidelines regarding antimicrobial use; thus, we chose the ISCAID recommendations [1,2,3,4,5] to assess the agreement of our study population practices with current clinical guidelines. We examined the theoretical use of antimicrobials for conditions where antimicrobials are not always required. These included uncomplicated feline upper respiratory tract disease, lower urinary tract disease in cats and dogs, acute gastroenteritis, superficial pyoderma, canine infectious tracheobronchitis (“kennel cough”), and elective surgeries. These conditions may not always be associated with a bacterial infection, may be self-limiting, can be associated with viruses, or have alternative therapies to the use of systemic antimicrobials. In general, the results showed that antimicrobials were over-used for each one of the conditions studied.

The treatment of superficial pyoderma for most dogs included at least one systemic antimicrobial alone (10% of respondents) or in combination with a topical product (79%). These findings do not follow ISCAID recommendations for treating superficial pyoderma [1]. Topical therapy (without co-administration of systemic antimicrobials) is encouraged, and the recommended approach for treating superficial pyoderma to avoid the emergence of multidrug-resistant infections by systemic antimicrobials. It has been found that methicillin-resistant *Staphylococcus pseudointermedius* (MRSP) is just as susceptible to topical treatments as methicillin-susceptible *Staphylococcus* (MSSP) and that aggressive topical therapy is precisely the best option to get a good cure [12]. Studies that have compared topical therapies with systemic ones report similar efficacy cures for MRSP [13]. Currently, there are novel products apart from the classic antiseptics with proven efficacy in dogs with pyoderma by MRSP [14]. One of the main risk factors that many studies associated with the resistance of MRSP in cases of pyoderma is the prior administration of antimicrobials, further emphasizing the need for prudent use of antimicrobials [15,16,17,18]. For example, in an epidemiological case study, the odds ratio (OR) of presenting MRSP versus MSSP infection was nine times higher in dogs treated with systemic antimicrobials 30 days before being referred to the hospital [19]. Apart from antimicrobial administration, another study reported that a higher proportion of MRSP was diagnosed in animals that received corticosteroids than those who did not [20]. Many of these animals probably did not respond to the antimicrobial therapy because they had some underlying problems such as atopy, as shown in another study [21].

In this study, 51/100 veterinarians stated they tried to determine the cause of pyoderma. However, only 24/100 asked pertinent questions (of the owner, themselves, and laboratory) to identify an underlying cause such as an allergy or mentioned possible endocrinopathies or ectoparasite infestations. These responses suggest a poor diagnostic plan for proper evaluation of any underlying primary comorbidities. Also, skin cytology was only performed in about 80–100% of the cases by 18 veterinarians. Although the diagnosis of pyoderma is based upon clinical signs and the presence of characteristic lesions, cytology from slide or tape impressions is mandatory when typical lesions (pustules) are not present, and for diagnosing co-infection with *Malassezia pachydermatis* or demodicosis, the former being a frequent occurrence in dogs with pyoderma [1]. Cytology is always recommended to confirm bacterial involvement as it has a 93% diagnostic sensitivity based on the presence of neutrophils and intracellular cocci [21,22].

In this study, the choices of suitable systemic antimicrobials for empirical therapy were following ISCAID recommendations, as half (50%) of the veterinarians chose cephalexin, a first-generation cephalosporin. First-tier drugs recommended by ISCAID include amoxicillin–clavulanate, first-generation cephalosporins, clindamycin or lincomycin, and trimethoprim–sulphonamides. However, our recent determination of the antimicrobial susceptibilities of 406 ear and skin isolates of *Staphylococcus pseudointermedius* from Colombian veterinary practices, found that only amikacin, amoxicillin–clavulanate, and ciprofloxacin had susceptibilities above 90%, whereas for cephalexin or trimethoprim–sulfadiazine, they were 81.6% and 57%, respectively [23]. In light of these findings, in our area, veterinarians are not using the “best” empirical antimicrobial for systemic therapy of superficial pyoderma.

Urinary tract infections (UTIs) are another leading reason for antimicrobial use, and we identified both deficiencies in the diagnosis and improper therapy in a wide range of situations. Prophylactic antimicrobial therapy for the prevention of cystitis in catheterized animals is never indicated [2,3], and yet, a majority (69%) of veterinarians prescribed a course of systemic antimicrobials for the duration of catheterization. The prevalence of bacteriuria in catheterized dogs and cats is high (10–55%) [24,25,26], but most of those cases represent subclinical bacteriuria, which does not require antimicrobial therapy. Antimicrobial administration does not prevent catheter-related UTI and should not be administered to these animals unless the bacterial infection is documented by urine culture [27]. Fifty-two veterinarians (52%) used the recommended method of cystocentesis for sample collection, and 48 (48%) used aerobic bacterial cultures and susceptibility testing to confirm the presence of infection and guide the choice of antimicrobial drugs. In addition, only two out of the six local diagnostic laboratories used by this cohort of veterinarians perform quantitative culture techniques and report the results as colony forming units (CFU)/mL, which is necessary to interpret if the level of bacterial growth is clinically significant. Veterinarians who did not use bacterial cultures relied on the combination of clinical signs (i.e., dysuria, pollakiuria, hematuria) and urinalysis (dipstick and cytological examination of the sediment) to diagnose sporadic bacterial cystitis (previously known as “uncomplicated UTIs”, [2,3]) in both dogs and cats. This method of diagnosis may be justified for female dogs with suspected sporadic bacterial cystitis. For cats, aerobic cultures should confirm the diagnosis in all cases due to the low likelihood of bacterial cystitis in animals with lower urinary tract signs, which are typically caused by feline idiopathic/interstitial cystitis or urolithiasis [2,3]. This diagnostic approach implies that many cats are being treated unnecessarily with antimicrobials for non-bacterial conditions such as feline idiopathic cystitis or urethral obstruction. Only five veterinarians (5%) used first-tier options (amoxicillin or trimethoprim–sulfonamides) to treat sporadic bacterial cystitis in dogs; fluoroquinolones (enrofloxacin, marbofloxacin, and ciprofloxacin) were first-line options for twenty-six veterinarians. These drugs are considered critically important in human medicine and should be reserved when veterinary use of first-tier options is not appropriate based on culture and susceptibility results or patient factors.

In a retrospective epidemiological study of UTIs in 1029 dogs classified with uncomplicated, complicated infections, or pyelonephritis, those with complicated infections that had previously received antimicrobials had a higher number of multiresistant bacteria (36% versus 21%), particularly in *E. coli* and *Staphylococcus* spp. [28]. In 21% of dogs with uncomplicated infections, that is, those that presented infections for the first time and had never been treated, there were multiresistant bacteria and no antimicrobial showed more than 90% susceptibility. Our recent study of urine cultures showed that *E. coli* was the most frequent pathogen isolated, isolated in 46.5% of 226 urine samples submitted for culture and susceptibility testing [23]. The antibiograms showed that only amikacin and florfenicol attained 100% susceptibility against *E. coli*, and for the two most frequently used antimicrobials (in this survey), susceptibilities were 69.2% for enrofloxacin and 67.7% for ampicillin. These findings imply that many dogs may not initially respond to any administered antimicrobial.

Regarding upper respiratory tract infections in dogs and cats, recommendations were made by the panel of experts of ISCAID [4]. Concerning cats, when the nasal discharges are serous, and there are no mucopurulent or purulent components, antimicrobial treatments should not be performed because a bacterial component is not normally complicating viral infections (herpesvirus and calicivirus). The ISCAID Working Group recommended considering antimicrobial treatment when fever, lethargy, or anorexia is present concurrently with mucopurulent nasal discharge. The optimal duration of treatment for a bacterial infection has not been established. The consensus is dosing for 7–10 days for an antimicrobial with good action against *Mycoplasma* spp. and *Chlamydia felis* such as doxycycline (first-line choice). If the response is adequate, therapy should continue for the drug for as long as there is progressive clinical improvement and at least 1 week past the clinical resolution of nasal discharge. However, the optimal duration of treatment is unknown and the recommendation was based on the Working Groups´ clinical experiences.

Like those of felines, in the dog, most of the upper respiratory signs´ etiologies are viral, and therefore, administration of antimicrobials is not indicated. Most dogs with clinical signs of “kennel cough” manifest a dry, raspy, acute onset of cough that ends in nausea, but is not accompanied by more serious clinical signs (fever, lethargy, inappetence, mucopurulent discharge) that warrant antimicrobial treatment. The expert panel recommended that if antimicrobial therapy is necessary, a drug with activity against *Bordetella bronchiseptica* and *Mycoplasma* spp. (such as a course of doxycycline for 7–10 days) could be used. Amoxicillin was considered an acceptable alternate first-line option when *Chlamydia felis* and *Mycoplasma* are not highly suspected. In the current study, antimicrobials were frequently prescribed for acute tracheobronchitis in dogs (46%). Also, the most frequent antimicrobials used were amoxicillin–clavulanate (11%). In a comparable survey in Belgium, antimicrobial drugs were frequently prescribed in cases of acute tracheobronchitis in dogs (68.8%) and upper respiratory tract disease in cats (43.7%) [11]. Nonetheless, antimicrobials are not indicated by ISCAID to treat acute uncomplicated tracheobronchitis in dogs [4]. Against these emerging standards, and because viral pathogens are the primary causal agents and the disease is usually self-limiting, antimicrobials are being overused for this condition.

Regarding therapies to treat acute diarrhea in dogs, our survey identified unnecessary and excessive use of antimicrobials. There were three accompanying conditions for which antimicrobials are not recommended [5], and yet, veterinarians prescribed antimicrobials for dehydration (36%), positive culture or PCR for bacterial enteropathogens (81%), and hemorrhage (74%). In a survey in the United Kingdom [29], 71% (263/371) of veterinarians prescribed antibiotics for diarrhea cases before resorting to first options based on nutritional management, deworming, probiotics, with or without some antidiarrheal agent [9]. There are still many myths regarding the management of uncomplicated acute diarrhea, but most cases are self-limiting and cured regardless of the treatment. One myth is that antibiotics must be administered in the case of an infection. The principle of good therapy lies in making a good diagnosis that includes a history, a physical exam with a rectal exam, and laboratory tests, that include a stool exam. However, the nature of diarrhea, particularly chronic diarrhea lasting more than three days, is such that a diagnosis is made based on response to treatment. Complicated cases are those in which the diarrhea is accompanied by sepsis—severe lethargy, fever, inflammatory leukogram (leukocytosis, leukopenia, deviation to the left, toxic neutrophils) in which bacterial infection endangers the life of the animal. For example, puppies with a *parvovirosis* should be treated with intravenous antibiotics.

Even in acute hemorrhagic gastroenteritis cases, better termed “acute hemorrhagic diarrheal syndrome” because the stomach is not involved and is typically caused by *Clostridium perfringens*, the use of antimicrobials has been contraindicated, and administration of probiotics has been shown to accelerate the clinical recovery [30]. Studies comparing the administration of antibiotics (amoxicillin/clavulanic acid) to symptomatic therapies in dogs with similar symptoms of acute hemorrhagic diarrhea (without signs of sepsis) of less than 3 days duration did not show differences in treatment efficacy, the severity of signs, or cure time [31]. However, veterinarians in our survey not only used a systemic antimicrobial drug for hemorrhagic diarrhea, but half (*n* = 50) used two antibiotics from different families. In a study evaluating the effect of metronidazole plus amoxicillin–clavulanic acid combined or separately in dogs with acute hemorrhagic diarrhea, there were no differences in hospitalization days or disease severity in the groups of treated animals [32]. Therefore, although bacteria can cause acute diarrhea in dogs, from all these studies it can be deduced that antimicrobials should not be administered as the first option, and reserved only for complicated cases (presence of fever, inflammatory leukogram) or when other therapies have not been effective.

Finally, perioperative antimicrobial prophylaxis should be based on evaluating the patient’s status (American Society of Anesthesiologists Classification, ASA) and the expected surgery (wound classification). As a general rule, low-risk patients ASA 1-2 with clean procedures such as elective ovariohysterectomies or castrations, do not require antimicrobial prophylaxis [33]. However, 86% of our cohort used systemic antimicrobials, particularly cephalexin (*n* = 52), for 7–10 days following castrations and ovariohysterectomies. This frequency of use in our cohort is much higher than a similar cohort in the United Kingdom, who used antimicrobials in only 32.1% of routine pre-scrotal castration and never used them in 31.1% of cases [34]. Similarly, most of our cohort prescribed a course of systemic antimicrobials, particularly the combination of metronidazole and spiramycin (Stomorgyl^®^ Boehringer-Ingelheim S.A, Bogota, Colombia), for dental cleaning procedures. The Danish Veterinary Association guidelines [33] recommend that neither gingivitis nor periodontitis require antimicrobial therapy and should be treated by removing dental plaques and tartar mechanically thorough dental cleaning. Antimicrobials should be reserved for cases of prominent swelling, pus, fever, and local lymphadenopathy.

## 5. Conclusions

This survey is the first investigation on antimicrobial use in companion animals in Colombia. The survey findings highlight the need to improve prudent antimicrobial use in every clinical condition evaluated by primary-care small animal practitioners. Such improvement should include doing more diagnostic tests (such as cultures and susceptibility testing), decreasing the use of antimicrobials, and selecting proper antimicrobial agents for empirical treatments. Lack of proper diagnostic work-up and improper case management was noted for urinary tract disease and superficial pyoderma. Cultures and susceptibility testing were insufficiently done for urinary tract infections, superficial pyoderma, in which cytology was also rarely performed, and elective surgeries, upper respiratory tract disease, and acute gastroenteritis for which antimicrobials were over-prescribed. A previous study of the antimicrobial susceptibility patterns of common bacterial in samples submitted to the veterinary diagnostic laboratory of the University of Antioquia also showed that, in the current survey, the choices of antimicrobials for empirical treatment of superficial pyoderma and urinary tract disease are inadequate due to high resistance. Our findings can be used to develop community-specific guidelines that influence veterinarians’ prescribing behavior and increase community awareness about when antimicrobial use is justified.

## Figures and Tables

**Figure 1 vetsci-08-00073-f001:**
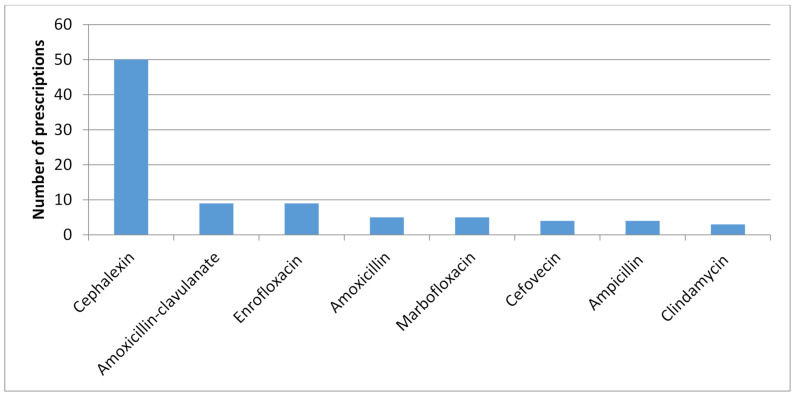
Histogram of the frequency of systemic antimicrobials agents initially prescribed by 100 Colombian veterinarians to treat canine superficial pyoderma.

**Figure 2 vetsci-08-00073-f002:**
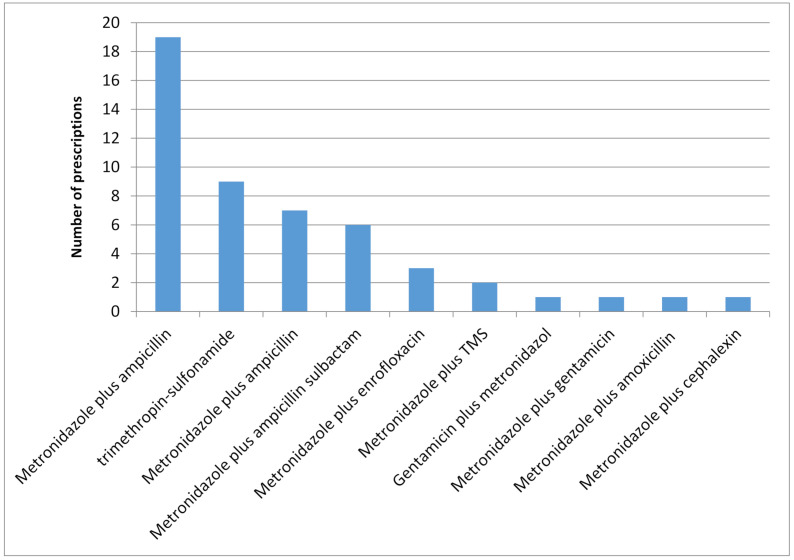
Histogram of the combinations of antimicrobials prescribed for bloody acute gastroenteritis (*n* = 50 veterinarians).

**Table 1 vetsci-08-00073-t001:** Demographic characteristics of 100 veterinarians interviewed for a survey of antimicrobial practices in 2020 in Medellin, Colombia, SA.

Characteristic	Distribution
Gender	55 males 45 females
Years in practice	
Between 3–10	45%
Between 10–20	35%
>20	20%
Type of practice	
Consulting office (no surgeries)	9%
Clinic	46%
Hospital (overnight care)	21%
Specialties or post-graduate studies	22%
Does your practice have written guidelines on the use of antimicrobials?	
Yes	11%
No	89%
Information sources to select antimicrobials (all that apply)	
Commercial literature	36%
International Veterinary Drug handbooks	49%
National Veterinary Drug Handbooks	22%
Scientific Journal	42%
Opinion of colleagues	47%
Textbooks	55%
Scientific Meetings	33%
What is your level of concern with regards to antimicrobial resistance (choose one):	
None	0%
Little	1%
Some	5%
Moderate	36%
Very high	58%
What is the percentage of generic antimicrobials that you tend to prescribe or think that your patients use? (choose one)	
0%	4%
25%	26%
50%	20%
75%	14%
100%	36%

**Table 2 vetsci-08-00073-t002:** Signs and frequency for which an infection of the upper respiratory tract in a cat would be prescribed with systemic antimicrobials (*n* = 100 veterinarians).

Signs	Frequency
Fever	84%
Lethargy	44%
Anorexia	27%
Seronasal secretions	64%
Mucopurulent secretions	97%
Conjunctivitis	51%
Sneezing	19%
Lacrimation (epiphora)	12%
Dehydration	63%

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
