# Peer review of "Antimicrobial Prescribing Practices in Dogs and Cats by Colombian Veterinarians in the City of Medellin"

_vetsci, 2021, doi:10.3390/vetsci8050073_

Round 1
Reviewer 1 Report
In this study the prescribing behavior of 100 companion animal veterinarians in the Columbian city of Medellin is surveyed with regard to current guidelines outlined by the International Society for Companion Animals on Infectious Diseases (ISCAID). A convenience sample of 100 primary-care veterinary practitioners was selected and a questionnaire was used to present 4 hypothetical clinical scenarios regarding prescription choices for systemic antimicrobials. It was found that antimicrobial prescribing behavior was not consistent with ‘best practice’ as outlined in the current guidelines. It was concluded that antimicrobial use could be improved by appropriate diagnostic steps leading to the choice of an appropriate antimicrobial drug. The discussion is well formulated although in the conclusion section there is reference to additional laboratory data not described in the manuscript. The manuscript would be improved if there was data available on the actual prescribing habits and antimicrobial use in the practices engaged. The title should reflect that the study was onducted in one city.
Author Response
RE: Corrections made in response to review comments for “Antimicrobial prescribing practices in dogs and cats by Colombian veterinarians in the city of Medellin”David A Gómez-Beltrán , David J Schaeffer, Duncan Ferguson, Laura K Monsalve and David Villar
We thank the reviewers for their insights and comments, and have edited the manuscript per their suggestions.
Reviewer 1
- Title: the location of the study, “Medellin” has now been included.
- Conclusion: the sentence alluding to our previous study has been modified to relate the results of that study with the present one.
A previous study Our recent studies of the antimicrobial susceptibility patterns of common bacterial species in samples submitted to the veterinary diagnostic laboratory of the University of Antioquia also showed that, in the current survey, the choice of antimicrobials for empirical treatment of superficial pyoderma and urinary tract disease were inadequate due to high resistance. [23].
- Conclusion: The manuscript would be improved if there was data available on the actual prescribing habits and antimicrobial use in the practices engaged. This is the first report on prescribing habits and antimicrobial use in Colombia.
Reviewer 2 Report
The Authors report a very interesting survey about the behavior of companion animal veterinarians about the antimicrobials prescription. I think that this paper can improve the awareness of clinicians of the antibiotics overuse/misuse and the importance to respect the correct guidelines to reduce the antimicrobial resistance threat. The paper appears to be well written and exhaustive. I suggest only to improve the results presentation reporting more graphics about other specific diseases in addition to superficial pyoderma; and to specify the route of administration, especially for gastroenteritidis cases.
Author Response
RE: Corrections made in response to review comments for “Antimicrobial prescribing practices in dogs and cats by Colombian veterinarians in the city of Medellin”David A Gómez-Beltrán , David J Schaeffer, Duncan Ferguson, Laura K Monsalve and David Villar
We thank the reviewers for their insights and comments, and have edited the manuscript per their suggestions.
Reviewer 2:
- A new graph has been included to show the most common combinations of antimicrobials to treat hemorrhagic diarrea. Results:”For hemorrhagic diarrhea, combinations of two antimicrobials were the preferred option by 50 of the participants (Figure 2)”.
- The route of administration was not asked in the questionnaire for gastroenteritis cases.
Reviewer 3 Report
General comments:
The manuscript “Antimicrobial prescribing practices in dogs and cats by Colombian veterinarians” provides information on drug administration in dogs by veterinarians. The paper's major strength, However, the article has far too many weaknesses eg the design did allow for replicability and data was not analysis in a meaningful manner. The Association between doctors (respondent) who said yes and no was not analysed. Many aspects of the result section are not drafted in a scientific manner as author did not provide sufficient statistic to support result findings.
Abstract
Poorly written with no actual results written in the form of frequencies and proportions.
Material and method
Provide ethical clearance number: Ethics approval was granted by the University of Antioquia Ethics Review Board
- Results: I suggest that authors must re-write result section and describe findings using percentages.
3.1. Demographics: result must be presented as a percentage proportion
Table 1: present result as a percentage proportion
Figure 1: include error bar
Conclusions: this must be based on findings and a generalization. Authors also in the conclusion segment, referred to the previous finding. This should not be found in conclusion. The current conclusion is poorly written and should be revised.
Author Response
RE: Corrections made in response to review comments for “Antimicrobial prescribing practices in dogs and cats by Colombian veterinarians in the city of Medellin”David A Gómez-Beltrán , David J Schaeffer, Duncan Ferguson, Laura K Monsalve and David Villar
We thank the reviewers for their insights and comments, and have edited the manuscript per their suggestions.
Reviewer 3
- Abstract:
- The numbers of veterinarian empirically prescribing a course of systemic antimicrobials for each scenerio…..is now expressed as percentages.
- The sentence: ”In superficial pyoderma cases, cytology was also rarely performed” has been replaced by: “In superficial pyoderma cases, cytology was performed in the following percent of cases: 0% (n=24), 20% (n=30), 40% (n=17), 60% (n=11), 80% (n=8), and 100% (n=10)”.
- The sentence: “In elective surgeries, upper respiratory tract disease, and acute gastroenteritis, antimicrobials were over-prescribed relative to emerging standards” has been replaced by: “Antimicrobials were over-prescribed relative to emerging standards for elective surgeries (86%), kennel cough (46%), and acute hemorrhagic diarrhea (50%).
- Percentages have been added to the sentence: “Critically important antimicrobials, such as fluoroquinolones, were applied commonly for superficial pyoderma (18%), kennel cough (12%), and lower urinary tract disease in dugs (20%) and cats (26%).
- Materials and Methods:
- Provide ethical clearance number: Ethics approval was granted by the University of Antioquia Ethics Review Board. The last sentence in materials and methods states: The Ethical Committee (“Comité de Bioetica para la Experimentación con humanos”) from the University of Antioquia on session number 20-113-918 approved this study.
- Results:
- Demographics: result must be presented as a percentage proportion. Table 1 now provides all results as percentage and so do the values throughout the results and discussion sections.
4) Conclusion:
- this must be based on findings and a generalization. Authors also in the conclusion segment, referred to the previous finding. This should not be found in conclusion. The current conclusion is poorly written and should be revised.
The sentence alluding to our previous study has been modified to relate the results of that study with the present one.
A previous study Our recent studies of the antimicrobial susceptibility patterns of common bacterial species in samples submitted to the veterinary diagnostic laboratory of the University of Antioquia also showed that, as in the current survey, the choices of antimicrobials for empirical treatment of superficial pyoderma and urinary tract disease were inadequate due to high resistance. [23].
Figure 1 include error bars This was a single survey. Error bars (standard deviations) show the variability of the data in replicate samples. Additional independent surveys are required to estimate the standard deviations.. To clarify the frequency distribution was for our single sample, we added “100” before “Colombian” to the figure legend.
The legend of Figure 1 was changed to: Figure 1. Histogram of the frequency of systemic antimicrobials agents initially prescribed by 100 Colombian veterinarians to treat canine superficial pyoderma
Similarly, the legend of Figure 2 was changed to: Figure 2. Histogram of the combinations of antimicrobials prescribed for bloody acute gastroenteritis (n=50 veterinarians).